# Cognitive load as a mediator in self-efficacy and English learning motivation among vocational college students

**Hui Zhang** *

Chengdu Agricultural College, Chengdu, Sichuan Province, China

* 16877867@qq.com

**Data Availability Statement:** https://www.openicpsr.org/openicpsr/project/204782/version/V1/view.

## Abstract

Chinese vocational colleges, under global and educational pressures, encounter challenges in IT integration for English teaching, which risks dampening student motivation due to heightened cognitive load. This study addresses the need to elucidate the mediating role of cognitive load in the relationship between self-efficacy and learning motivation among these students. By integrating Social Cognitive Theory(SCT) and Cognitive Load Theory(CLT), the research employs a structural equation model to analyze survey data, revealing that self-efficacy positively impacts deep and achievement motivation and negatively influences cognitive load, which in turn affects learning motivation. These insights underscore the importance of fostering self-efficacy and managing cognitive load to enhance vocational students' English learning motivation, offering valuable guidance for educational practices and curriculum development in the face of global challenges.

## Introduction

In the context of rapid globalization and educational reforms, vocational colleges face increasing pressure to enhance teaching quality and expand students' abilities. English education, as a crucial component of global competitiveness, not only reflects students' international competencies but also contributes to the elevation of educational standards. To navigate these challenges, Chinese vocational colleges have embraced the digital era by adopting information technology to innovate learning resources and environments. For example, they have adopted online course platforms such as China University MOOC, virtual classrooms like Tencent Meeting, interactive e-textbooks with multimedia content through U-Campus APP, mobile learning apps like Duolingo, and collaborative tools such as WeChat for group discussions. Additionally, some institutions have integrated AI-assisted tools like Grammarly and Doubao, and learning management systems (LMS) like Xuexitong.

While these technological advancements have introduced a wealth of new tools and opportunities, they also present complexities that can potentially hinder student motivation. The digital era is characterized by an abundance of technological tools and resources that promise to enhance personalized learning experiences [1]. However, despite these advancements, there

**Funding:** The author(s) received no specific funding for this work.

**Competing interests:** The authors have declared that no competing interests exist.

is a notable trend of decreased initiative and enthusiasm among students, particularly in higher vocational education [2]. Recent studies indicate that although access to digital resources has increased, the actual engagement and active participation of students have not necessarily followed suit [3]. This discrepancy highlights the need to address the motivational challenges posed by the digital age, especially by considering individual differences in self-efficacy, which may influence how students interact with and benefit from these technologies.

To tackle the issue of weak learning motivation, existing literature has explored various pedagogical approaches, including gamification [4], the Community of Inquiry (CoI) framework [3], and the integration of augmented reality (AR) and digital games [5]. Despite these efforts, there is a significant gap in the literature regarding the role of differentiated instructional strategies based on individual student characteristics, particularly in the context of English language learning. Self-efficacy, as a critical antecedent of learning motivation, has been widely studied as an outcome of different teaching methods [6]. However, the application of tiered instructional strategies tailored to self-efficacy differences, which could be instrumental in enhancing motivation, remains underexplored. Given the personalization afforded by the digital era, it is essential to consider how variations in self-efficacy among students can inform and optimize instructional designs.

Furthermore, the cognitive load experienced by students in the digital age poses a significant challenge. The complexity and quantity of information can overwhelm learners, thereby diminishing their motivation [7]. High cognitive load, characterized by excessive demand on working memory, is a common occurrence in higher vocational education, where students are required to process vast amounts of technical and specialized knowledge [8]. Research indicates that high cognitive load can lead to decreased engagement and poor academic performance [9]. Therefore, managing cognitive load effectively is crucial for sustaining or improving learning motivation.

While previous research has delved into the antecedents of cognitive load, such as task difficulty and prior knowledge [10], there is a paucity of studies that examine how individual differences, particularly in self-efficacy, influence the perception and management of cognitive load. The distinctive feature of personalized learning in the digital era necessitates a closer look at how self-efficacy variances affect cognitive load, an area that remains largely uncharted. It is hypothesized in this study that students with varying levels of self-efficacy will perceive the same learning content differently in terms of cognitive load, thereby influencing their overall motivation.

Moreover, the construct of learning motivation is multifaceted, encompassing external, intrinsic, and achievement-oriented motivations [11]. In the digital context of higher vocational English learning, the pathways through which these different types of motivations develop may vary. Existing research has not provided clear answers regarding the specific mechanisms through which these motivations form and interact, highlighting the need for further investigation. Understanding the nuances of motivation development is crucial for designing effective interventions that can leverage the benefits of digital technologies while mitigating their potential drawbacks, ultimately fostering a more motivated and engaged student body.

To address these gaps, this study aims to investigate the determinants of learning motivation, with a specific focus on the influence of self-efficacy and the mediating role of cognitive load. By examining these factors in the context of Chinese vocational college students, this research seeks to provide empirical support for the field of Chinese vocational education. Additionally, the findings will offer concrete guidance for international educators in designing and implementing cross-cultural teaching programs. The research outcomes are expected to directly impact educational practices, particularly in developing vocational education curricula

and teaching methods that are adaptable to global challenges, thereby assisting educators in more effectively promoting students' English learning and professional development in a multicultural context.

Theoretically, this study is grounded in social cognitive theory, which highlights the pivotal role of self-efficacy in shaping individuals' motivation and behavior. By focusing on self-efficacy, this research seeks to enhance our understanding of how individual differences can be leveraged to improve learning outcomes, thereby contributing to the enrichment of the social cognitive theory. Additionally, this investigation aims to extend cognitive load theory by examining how self-efficacy influences cognitive load. This contribution is crucial as it introducing the concept of subjective cognitive load. Traditional cognitive load theory primarily focuses on the objective cognitive processing, neglecting the learners' subjective feelings. Considering cognitive load as a sensation allows for a more comprehensive understanding of the cognitive load phenomena in the learning process. Furthermore, this study aims to contribute to self-determination theory by exploring the interplay between self-efficacy and cognitive load in shaping different types of motivation, providing a more comprehensive model of motivation in the digital learning environment. These contributions are significant as they offer a robust theoretical foundation for addressing the motivational and cognitive challenges faced by students in the digital age, facilitating the development of more personalized and effective educational practices.

## Literature review

### Learning motivation

Learning motivation refers to the intrinsic drive that individuals exhibit during the learning process, encompassing factors such as interest, goal orientation, and a sense of achievement [12]. Despite the rich learning resources provided by digital technologies, numerous studies indicate that students in the digital age face a decline in learning motivation. Key factors contributing to this decline include attention dispersion, information overload, technology dependence, and self-management difficulties [7,13–15]. Attention dispersion, caused by multitasking and frequent interruptions, hinders students' ability to concentrate, negatively impacting learning outcomes [7]. Information overload, characterized by the vast amount of online information, makes it difficult for students to filter out valuable content, leading to reduced learning efficiency [13]. Technology dependence can lead to a lack of independent thinking and problem-solving skills [14]. Additionally, self-management difficulties, particularly in autonomous learning, are common due to the high levels of self-regulation required [15]. These factors highlight the critical role of self-efficacy in mitigating the negative effects of high cognitive load on learning motivation.

The concept of learning motivation is multifaceted, including intrinsic motivation, extrinsic motivation, and achievement motivation [16]. Intrinsic motivation stems from personal interest and enjoyment in the learning process, while extrinsic motivation is driven by external rewards or consequences, such as grades or praise. Achievement motivation involves the desire to perform tasks optimally and improve one's capabilities [3]. Extensive research has explored these categories. The Self-Determination Theory, which differentiates between various forms of intrinsic and extrinsic motivation, has been a significant framework in this exploration. [17] studies have investigated the effects of external rewards on intrinsic motivation, [18], the concept of achievement motivation has been explored within the framework of self-regulated learning [19]. However, these studies lack an in-depth exploration of the mechanisms underlying the formation of different types of motivation. Understanding these distinct motivational

components is essential for creating a supportive learning environment that meets the diverse needs of students.

## Cognitive load

Cognitive load refers to the working memory burden experienced during the learning process [20]. Numerous studies have shown that cognitive load significantly influences learning motivation. High cognitive load can lead to feelings of stress and frustration, thereby reducing learning motivation [21]. These studies have explored how different technological interventions can either alleviate or exacerbate cognitive load, subsequently affecting learners' motivation [22]. In the digital age, high cognitive load negatively impacts students' learning motivation. For example, complex multimedia environments can confuse and stress students, leading to a decrease in motivation [22]. Information overload further exacerbates cognitive load [13], and the technological distractions, and multitasking all contribute to this high cognitive load [23]. Therefore, reducing cognitive load in digital learning environments is crucial for enhancing the learning motivation of vocational college students.

Cognitive load can be categorized into intrinsic cognitive load, extraneous cognitive load, and germane cognitive load [24]. Intrinsic cognitive load is determined by the nature of the learning material, related to its complexity and structure. Extraneous cognitive load is induced by instructional design, associated with teaching methods and presentation styles. Germane cognitive load arises from the learner's active construction of knowledge, linked to cognitive strategies and effort. Subsequent research has explored management strategies for these types of cognitive load. Discussions have centered on how cognitive loads impact learners' processes and outcomes, and how instructional design can be utilized to manage these loads effectively [25–27]. Various studies have measured intrinsic, extraneous, and germane cognitive loads, examining their respective impacts on learning outcomes [28,29]. However, given the high cognitive load in the digital age, objectively managing cognitive load has limitations. Cognitive load is not only an objective measure but also a subjective experience [30] Subjective cognitive load refers to the perceived level of cognitive burden during task performance [30]. The importance of considering this subjectivity when assessing cognitive load has been underscored [31]. Therefore, exploring the impact of different contexts on learners' subjective experiences of cognitive load could provide new pathways to address the high cognitive load faced by vocational college students in the digital age.

Recent studies have highlighted the importance of self-efficacy in regulating students' subjective cognitive load. Research indicates that students with high self-efficacy can more effectively manage cognitive load [32], reducing their subjective perception of it. Furthermore, it has been discovered that teachers who provide differentiated support based on students' self-efficacy can reduce subjective cognitive load and enhance learning motivation [33]. These findings suggest that students with different levels of self-efficacy may perceive different levels of subjective cognitive load when encountering the same learning materials and instructional designs. Therefore, tiered instruction based on self-efficacy differences can help influence subjective cognitive load and, consequently, learning motivation.

## Self-efficacy

Self-efficacy, a core concept of Social Cognitive Theory (SCT), refers to the belief in one's ability to perform tasks necessary to achieve desired outcomes [34]. This construct is pivotal in shaping behavior and motivation, influencing goal setting, effort, and persistence [35]. Students with high self-efficacy are more likely to persist in the face of challenges, thereby enhancing their learning motivation. Studies have demonstrated a significant positive correlation

between self-efficacy and learning motivation [36–38]. Studies in the digital age similarly support the positive impact of self-efficacy on learning motivation [39,40]. Therefore, differentiated instruction strategies, based on students' self-efficacy levels, provide appropriate support and challenges [41], thus meeting the diverse needs of students and enhancing learning motivation [42].

Given that the high cognitive load in the digital age is an objective reality, it highlights the importance of exploring contextual factors that influence the subjective perception of cognitive load as a viable approach to improving learning motivation. Previous literature suggests that differentiated instruction based on self-efficacy differences may be a potential contextual factor affecting students' subjective cognitive load [5]. However, this hypothesis requires further empirical validation, as most studies on cognitive load and self-efficacy have focused on the impact of cognitive load on self-efficacy rather than the reverse [5,8,21,43,44]. While these studies provide valuable insights into the relationship between cognitive load and self-efficacy, there is a need for more research focusing on whether self-efficacy influences subjective cognitive load. Understanding how to manage cognitive load from the perspective of students' subjective experiences is essential for enhancing learning motivation and requires further empirical investigation.

Hypotheses

Self-efficacy and english learning motivation

Self-efficacy, a core concept of Social Cognitive Theory (SCT) is refer to the belief in one's ability to perform tasks necessary to achieve desired outcomes [45]. This construct is pivotal in shaping behavior and motivation, influencing goal setting, effort, and persistence [46]. In educational settings, high self-efficacy fosters a strong drive to learn, as individuals are more likely to tackle challenging tasks and persevere through obstacles. The theoretical mechanism linking self-efficacy to learning motivation is supported by research across various disciplines. Studies have shown that game-based learning, which enhances self-efficacy, significantly improves students' motivation and performance [35]. This finding is consistent with SCT's emphasis on mastery experiences as a means to bolster self-efficacy. Similarly, Research has found that integrating flipped classrooms with MOOCs and game-based learning can enhance motivation, particularly for students with lower self-confidence [47], suggesting that such interventions can effectively increase self-efficacy, and subsequently, their motivation to learn.

In the context of English language learning among Chinese vocational students, SCT provides a framework for understanding the impact of self-efficacy on motivation. Chen et al. highlighted the role of self-efficacy in engaging learners through digital strategies, which is particularly relevant for English language acquisition [48]. Admiraal et al. showed that autonomy-supportive teaching practices, which likely enhance self-efficacy, positively influence students' learning motivation [21].

These studies collectively support the theoretical logic that self-efficacy is a critical determinant of learning motivation, with educational interventions that enhance self-efficacy having the potential to significantly improve students' engagement and performance in English language learning. Based on these ideas, hypothesis one is as follows:

H1. Self-efficacy positively influences English learning motivation.

According to the Expectancy-Value Theory proposed by Eccles and Wigfield, individuals engage in activities based on their expectation of success and the value they place on those activities [49]. High self-efficacy enhances an individual's expectation of success, making them more likely to pursue goals associated with external rewards, as they believe they can achieve these goals. In the context of English learning in vocational colleges, students with high self-efficacy are more confident in their ability to learn English, which makes them more proactive

in pursuing external rewards such as good grades, teacher praise, and other forms of recognition. These students are also more likely to set specific learning goals, such as passing English exams, obtaining scholarships, or securing better job opportunities, because they believe they can achieve these through their efforts. The confidence in their abilities makes these external rewards more attractive, thereby further enhancing their extrinsic motivation. Empirical research supports this notion, finding that self-efficacy is significantly we put forth the following hypothesis.

H1a. Self-efficacy positively influences extrinsic motivation.

According to Self-Determination Theory (SDT), autonomy, competence, and relatedness are fundamental psychological needs that drive human behavior [17]. When individuals feel a high level of competence in an activity, such as having high self-efficacy, they experience greater autonomy and control, which in turn enhances their intrinsic motivation. In the context of English learning in vocational colleges, students with high self-efficacy believe in their ability to learn English well, leading to increased confidence and comfort during the learning process. This confidence makes them more likely to find enjoyment and satisfaction in English learning, as they believe they can master new knowledge and skills, thereby further enhancing their intrinsic interest and motivation. Empirical research supports this notion, with studies finding that self-efficacy has a significant impact on intrinsic motivation [50]. Additionally, high-creative self-efficacy postgraduate students develop higher levels of intrinsic motivation after receiving developmental feedback from academic supervisors compared to those with low creative self-efficacy [51]. Furthermore, among inclusive education teachers, the dimensions of self-efficacy, including inclusive teaching efficacy, collaborative efficacy, and behavioral management efficacy, show the strongest predictive effects on intrinsic motivation [52]. Creative self-efficacy indirectly influences graduate student creativity through its effect on intrinsic motivation [53], and the relationship between self-efficacy and creativity is fully mediated by intrinsic motivation [54]. These findings collectively support the hypothesis that self-efficacy positively influences the intrinsic motivation of vocational college students in English learning. Based on the above analysis, this study hypothesizes that:

H1b. Self-efficacy positively influences intrinsic motivation.

Expectancy-value theory posits that individuals choose to engage in an activity because they expect to successfully complete it (expectancy) and perceive it as valuable (value). According to this theory, those with high self-efficacy typically have higher success expectancies, thereby enhancing their achievement motivation. Specifically, studies have noted that students with high self-efficacy are more likely to set challenging goals and persist in the face of difficulties [55], aligning with the "expectancy" component of expectancy-value theory, as these students have stronger confidence in their ability to accomplish tasks. When individuals are confident in their abilities (i.e., have high self-efficacy), they have higher expectations of success, and this high expectancy, combined with the value of the task, further strengthens their achievement motivation [56]. In the context of English learning in vocational colleges, high self-efficacy positively influences achievement motivation because students with high self-efficacy believe in their ability to learn English well (high success expectancy) and recognize the importance of mastering English for their future career development (high task value), making them more likely to exhibit strong achievement motivation as they not only anticipate success but also view the goal as worthwhile. Based on the above analysis, this study hypothesizes that:

H1c. Self-efficacy positively influences achievement motivation.

## Self-efficacy and cognitive load

Cognitive load refers to the mental effort required to process information during learning, influenced by task complexity and learner's prior knowledge [20]. The Self-Efficacy and Cognitive Load Theory (SECLT), suggests that self-efficacious learners are more likely to engage in effective learning strategies, thereby reducing cognitive load by managing the demands on working memory [57]. This theoretical framework posits that the activation of metacognitive strategies, such as goal setting, progress monitoring, and strategy adjustment, enables learners to organize and integrate new information with existing knowledge, preventing cognitive overload and enhancing learning effectiveness. Hwang et al. demonstrated that peer assessment-based game development approaches can improve learning achievements, motivations, and problem-solving skills [58], aligning with the SECLT's emphasis on self-regulation and goal setting. Similarly, Wang et al. showed that elaborated feedback can enhance self-regulated learning and performance, indicating that feedback can support self-efficacious learners in managing cognitive load [59]. These findings underscore the importance of self-efficacy in optimizing cognitive resource management across diverse educational settings.

In the specific context of English language learning among Chinese vocational college students, the SECLT provides a valuable lens through which to understand the impact of self-efficacy on cognitive load. Self-efficacy plays a pivotal role in creating a conducive learning environment, as highlighted by research in the field [60]. This factor is particularly important for reducing cognitive load and enhancing language acquisition. A more recent study further underscores the influence of self-efficacy on language learning motivation [61]. It suggests that students with a strong sense of self-efficacy are more inclined to adopt strategies that alleviate cognitive load. These strategies include setting ambitious goals and efficiently tracking their learning progress, which are instrumental in the language learning process. Consequently, educators have the opportunity to craft learning environments that foster self-efficacy, which in turn can optimize the allocation of cognitive resources and facilitate language acquisition. Based on the above analysis, this study hypothesizes that:

H2. Self-efficacy negatively influences cognitive load among vocational college students.

## Cognitive load and english learning motivation

Cognitive Load Theory is a framework that explains how the cognitive demands of learning tasks can impact the efficiency of information processing and, consequently, learning outcomes [24], which posits that the human cognitive system has a limited capacity for processing information, and when this capacity is exceeded, cognitive overload occurs, leading to reduced learning effectiveness [62]. This theory has been applied across various educational contexts to understand the cognitive demands placed on learners and to inform instructional design strategies that aim to optimize learning. For example, Hwang, Chien, and Li suggests that by managing cognitive load, educators can create learning environments that foster motivation and engagement [35]. Similarly, Huang et al. demonstrated that spherical video-based virtual reality (SVVR) learning systems can improve learning motivation by providing an immersive and engaging experience that potentially reduces cognitive load [22], thereby enhancing students' willingness to engage with the learning material.

Turning to the specific context of English language learning among vocational college students in China, CLT offers insights into how cognitive load can influence learning motivation. High cognitive load in language learning can arise from complex grammar structures, unfamiliar vocabulary, and the need for rapid processing of auditory input, all of which can challenge students' cognitive capacity [63]. When students perceive these demands as

overwhelming, their self-efficacy may decrease, leading to reduced motivation to continue learning [64]. These finding underscores the importance of considering cognitive load in the design of English language teaching materials and strategies for vocational college students in China. Consequently, this study hypothesizes that:

H3. Cognitive load negatively influences English learning motivation among vocational college students.

According to Cognitive Load Theory (CLT), high cognitive load consumes limited cognitive resources, negatively impacting learning outcomes and motivation. In the context of English learning in vocational colleges, complex grammatical structures, extensive vocabulary, and listening exercises increase intrinsic cognitive load, while unclear instructional materials and complex information presentation add extraneous cognitive load. When students feel their cognitive resources are insufficient, they struggle to focus on external rewards such as grades and teacher praise, as they must concentrate on the immediate learning tasks. High cognitive load can also induce anxiety and frustration, further reducing interest in external rewards. Empirical research supports this, showing that high cognitive load reduces task focus and lowers extrinsic motivation [20]. Consequently, this study hypothesizes that:

H3a. Cognitive load negatively influences extrinsic motivation among vocational college students.

High cognitive load makes students feel resource-constrained, hindering their ability to focus on tasks and achieve optimal performance. In the context of English learning in vocational colleges, complex grammatical structures, extensive vocabulary, and listening exercises increase intrinsic cognitive load, while unclear instructional materials and complex information presentation add extraneous cognitive load. When students feel their cognitive resources are insufficient, they struggle to focus on the immediate learning tasks, which can reduce their intrinsic interest and enjoyment in the learning process. High cognitive load can also induce anxiety and frustration, further diminishing their intrinsic motivation. Empirical research supports this notion, with studies indicating that high cognitive load reduces students' focus on tasks and thus lowers their intrinsic motivation [20]. Therefore, this study hypothesizes that:

H3b. Cognitive load negatively influences intrinsic motivation among vocational college students.

In the digital age, vocational college students learning English face additional cognitive load from using various digital tools and resources. Multitasking, information overload, and technical issues increase extraneous cognitive load, making it harder for students to concentrate on improving their abilities and task quality, thereby reducing their achievement motivation. Consequently, this study hypothesizes that:

H3c. Cognitive load negatively influences achievement motivation among vocational college students.

## The mediating effects of cognitive load

Self-efficacy has been extensively studied for its influence on learning motivation. Empirical evidence suggests that self-efficacy positively influences learning motivation, as students with high self-efficacy are more likely to engage in challenging tasks and persist in the face of difficulties [43]. Conversely, self-efficacy has also been linked to cognitive load, with higher self-efficacy potentially reducing the perceived cognitive demands of a task [60]. The negative

impact of cognitive load on learning motivation is well-documented, with studies indicating that excessive cognitive demands can lead to decreased motivation and engagement [25]. This relationship is particularly relevant in the context of English language learning for vocational students, where high cognitive load can hinder the development of language skills and strategies [60].

The current study integrates the SCT and CLT. The integrated framework suggests that self-efficacy, a psychological construct central to SCT, can influence learning motivation by modulating the cognitive load experienced by learners. High self-efficacy can reduce cognitive load by enhancing students' ability to manage and regulate their cognitive resources, thereby facilitating more efficient learning and increasing motivation. This mediating role of cognitive load is supported by findings from studies [58,59], which show that interventions that increase self-efficacy also reduce cognitive load and improve learning outcomes. Based on the above discussion, this study posits the following hypotheses:

H4. Cognitive load mediates the relationship between self-efficacy and learning motivation

Self-efficacy, or the belief in one's ability to succeed in specific situations, has been shown to positively influence learning motivation. Students with high self-efficacy are more likely to engage in challenging tasks and persist despite difficulties[46]. In the context of extrinsic motivation, which is driven by external rewards such as grades and praise, high self-efficacy can reduce the perceived cognitive demands of a task, thereby lowering cognitive load. When cognitive load is reduced, students have more cognitive resources available to focus on external rewards, enhancing their extrinsic motivation. Empirical evidence supports this relationship, with studies showing that interventions aimed at increasing self-efficacy also reduce cognitive load and improve learning outcomes [65]. Therefore, it is hypothesized that cognitive load mediates the relationship between self-efficacy and extrinsic motivation.

H4a. Cognitive load mediates the relationship between self-efficacy and extrinsic motivation

Intrinsic motivation, which arises from the inherent interest and enjoyment in the learning process itself, is also influenced by self-efficacy. Students with high self-efficacy are more likely to find the learning process enjoyable and engaging because they feel capable of mastering the material. High self-efficacy can reduce the perceived cognitive demands of a task, thus lowering cognitive load. When cognitive load is lower, students can more easily focus on the intrinsic aspects of the learning process, such as curiosity and interest, leading to higher intrinsic motivation. Research indicates that reducing cognitive load through self-efficacy-enhancing interventions can improve intrinsic motivation [66]. Therefore, it is hypothesized that cognitive load mediates the relationship between self-efficacy and intrinsic motivation.

H4b. Cognitive load mediates the relationship between self-efficacy and intrinsic motivation

Achievement motivation, the desire to accomplish tasks to the best of one's ability and to improve competence, is closely linked to self-efficacy. Students with high self-efficacy are more likely to set and achieve high goals, persist in the face of challenges, and seek to improve their skills [67]. High self-efficacy can reduce the perceived cognitive demands of a task, thereby lowering cognitive load [68]. When cognitive load is reduced, students can allocate more cognitive resources to the task at hand, allowing them to focus on achieving their goals and improving their performance. This, in turn, enhances their achievement motivation. Recent studies have shown that interventions that increase self-efficacy also reduce cognitive load and improve learning outcomes, supporting the mediating role of cognitive load [65]. Therefore, it is hypothesized that cognitive load mediates the relationship between self-efficacy and achievement motivation.

H4c. Cognitive load mediates the relationship between self-efficacy and achievement motivation

## Research method

Adhering to the ethical guidelines and standards for research with human subjects as per Chinese laws and regulations, the study design underwent a thorough review and received approval from the relevant ethics review board. The conduct of the research prioritized the rights, safety, and well-being of all participants. All participants provided informed consent after being given a detailed account of the study's objectives, procedures, potential risks, benefits, and their right to withdraw without any adverse consequences. Documentation of consent was secured in alignment with principles promoting voluntary participation and confidentiality.

The investigation delves into the impact of self-efficacy on English learning motivation among students in vocational colleges in China, particularly examining the mediating effect of cognitive load. By referencing the existing body of literature, a preliminary research model is formulated to delineate the interplay between these psychological constructs. Fig 1 presents the conceptual model, where self-efficacy is recognized as the independent variable; learning motivation is the dependent variable; and cognitive load serves as the mediator variable.

This study employs a series of scales to measure self-efficacy, cognitive load, and learning motivation among English language learners in higher vocational colleges in China. The scales were adapted and validated to ensure their relevance and reliability within the research context. The English learning self-efficacy scale, originally developed by Zhang Risheng and Yuan Limin [69], was modified to include four subscales: "sense of English learning ability," "confidence in achieving goals," "ability to face setbacks," and "ability to overcome difficulties." The scale consists of 15 items, each rated on a 5-point Likert scale from "very disagree" *(1)* to "very agree" (5). The Cognitive Load Scale for English Learning is based on the NASA-TLX scale developed by the National Aeronautics and Space Administration (NASA). This scale assesses six dimensions: mental demand, physical demand, time demand, effort level, performance level, and frustration level. The 9-point Likert scoring system ranges from "very inconsistent" (1) to "very consistent" (9). The Learning Motivation Scale is adapted from the Learning Process Questionnaire by Zhang Risheng and Yuan Limin [69], with further revisions by Lei Li, Hou Zhijin, and Bai Xuejun [70]. The scale comprises 16 items across three subscales: extrinsic motivation, deep motivation, and achievement motivation. Rated on a 9-point Likert scale.

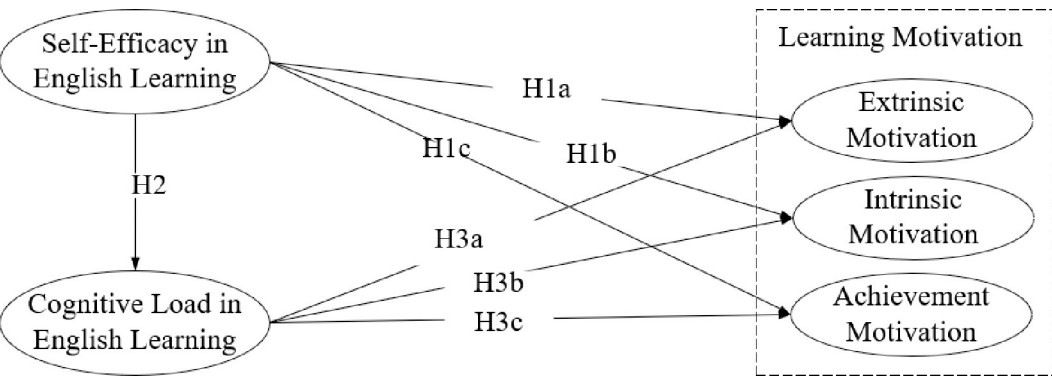

**Fig 1. Conceptual model.**

For the analysis of the structural equation model, Mplus 8.7 software was utilized, providing a robust platform for examining the relationships between the latent and observed variables within the study's theoretical framework.

## Research findings

### Respondent Profile

The study's sample was meticulously selected to ensure a representative cross-section of the student body. A total of 200 college students were randomly chosen from XX University and XX Vocational College, with an equal distribution of 100 students from each institution. This balanced approach aimed to capture a diverse range of academic experiences and learning environments. The demographic composition of the sample included 75 male and 125 female students, reflecting the gender distribution within the institutions. To provide a more nuanced understanding of the participants' academic backgrounds, the sample was stratified by academic major, encompassing a variety of fields such as engineering, business, humanities, and applied sciences. Additionally, the sample was divided into different years of study, from first-year to final-year students, to account for the potential impact of academic progression on self-efficacy and motivation. English proficiency levels were assessed using standardized tests and self-reported language skills. Participants were categorized into three levels of English proficiency: beginner, intermediate, and advanced, based on their scores and self-assessment. This classification allowed for a more detailed analysis of how cognitive load mediates the relationship between self-efficacy and English learning motivation across different proficiency levels.

Data collection was conducted via a comprehensive questionnaire survey, which was administered through an online platform to maximize accessibility and participation. The survey was open from March to April 2023, aligning with the academic calendar to ensure high engagement from students. Out of the 200 questionnaires distributed, 168 were returned, and after a thorough review, all were deemed valid, resulting in an effective recovery rate of 84%. The high response rate attests to the relevance and interest of the topic among the student population and contributes to the reliability of the study's findings.

### Confirmatory factor analysis

To ensure the validity and reliability of the measurement scales, the study conducted confirmatory factor analysis (CFA) for three key constructs: English learning self-efficacy, cognitive load, and learning motivation. The analysis aimed to validate the modified scales and their respective subscales, ensuring that each item loaded onto the expected construct and that the constructs were appropriately correlated.

The CFA results for the three constructs—English learning self-efficacy, cognitive load, and learning motivation—were all assessed to ensure the scales' reliability and validity. The Self-Efficacy Scale for English Learning, detailed in Table 1, showed a good fit with indices of CFI = 0.942, TLI = 0.928, RMSEA = 0.068, and SRMR = 0.054, with a p-value = 0.065 for the goodness-of-fit test. The Cognitive Load in English Learning scale, presented in Table 2, demonstrated an exceptional fit with a CFI = 1.000, TLI = 1.021, RMSEA = 0.000, and SRMR = 0.027, and a p-value = 0.865 for the exact goodness-of-fit test. Lastly, the English

**Table 1. Fitting goodness test of english learning self-efficacy.**

| CFI | TLI | RMSEA | P-Value of Goodness-of-Fit Test | SRMR |
|-----|-----|-------|--------------------------------|------|
| 0.942 | 0.928 | 0.068 | 0.065 | 0.054 |

**Table 2. Fitting goodness test of cognitive load in english learning.**

| CFI | TLI | RMSEA | P-Value of Goodness-of-Fit Test | SRMR |
|-----|-----|-------|--------------------------------|------|
| 1.000 | 1.021 | 0.000 | 0.865 | 0.027 |

Learning Motivation scale, as shown in Table 3, indicated a good fit with a CFI = 0.934, TLI = 0.915, RMSEA = 0.05, and SRMR = 0.077, with a p-value = 0.482 for the exact goodness-of-fit test. All scales met the criteria for a good model fit, confirming their appropriateness for the study.

## Structural equation model

The study utilized a structural equation model (SEM) to examine the proposed causal links among the variables. SEM integrates features of both multiple regression and factor analysis, allowing for the simultaneous estimation of a network of relationships among multiple variables [71]. According to the research conceptual model in Fig 1, the equation model is:

$$\begin{cases} \eta_1 = \beta_{14}\eta_4 + \gamma_1\xi + \zeta_1 \\ \eta_2 = \beta_{24}\eta_4 + \gamma_2\xi + \zeta_2 \\ \eta_3 = \beta_{34}\eta_4 + \gamma_3\xi + \zeta_3 \\ \eta_4 = \gamma_4\xi + \zeta_4 \end{cases} \tag{Eq1}$$

$$\text{Its matrix form is :} \begin{bmatrix} \eta_1 \\ \eta_2 \\ \eta_3 \\ \eta_4 \end{bmatrix} = \begin{bmatrix} \beta_{14} \\ \beta_{24} \\ \beta_{34} \\ 0 \end{bmatrix} \eta_4 + \begin{bmatrix} \gamma_1 \\ \gamma_2 \\ \gamma_3 \\ \gamma_4 \end{bmatrix} \xi + \begin{bmatrix} \zeta_1 \\ \zeta_2 \\ \zeta_3 \\ \zeta_4 \end{bmatrix} \tag{Eq2}$$

In the equation, $\xi$ represents the exogenous variables, i.e., the self-efficacy of English learning among college students; $\eta$ represents the endogenous variables, including the extrinsic motivation $\eta_1$, deep motivation $\eta_2$, achievement motivation $\eta_3$ and the cognitive load of college students' English learning $\eta_4$. $\gamma$ represents the path coefficient between exogenous variables and endogenous variables, and $\beta$ represents the path coefficient between endogenous variables, and $\zeta$ is the random disturbance term of endogenous variables. The research uses the robust maximum likelihood estimator (MLR) to estimate the significance level of each path system and to test the mediating effect of cognitive load in English learning.

The computational result of the model is shown in Fig 2. Among them, English learning self-efficacy significantly affects intrinsic motivation ($\beta = 0.573$, $p < 0.001$) and achievement motivation ($\beta = 0.491$, $p < 0.05$), so hypothesis H1band hypothesis h1c are supported. English learning self-efficacy significantly negatively affects cognitive load in English learning ($\beta = 0.419$, $p < 0.05$), so hypothesis H2 is supported. Cognitive load in English learning significantly negatively affects extrinsic motivation ($\beta = -0.772$, $p < 0.001$), so hypothesisH3a is supported. However, English learning self-efficacy has no significant effect on extrinsic motivation ($\beta = 0.054$, $p > 0.05$), and cognitive load in English learning has no significant effect on intrinsic

**Table 3. Fitting goodness test of english learning motivation.**

| CFI | TLI | RMSEA | P-Value of Goodness-of-Fit Test | SRMR |
|-----|-----|-------|--------------------------------|------|
| 0.934 | 0.915 | 0.05 | 0.482 | 0.077 |

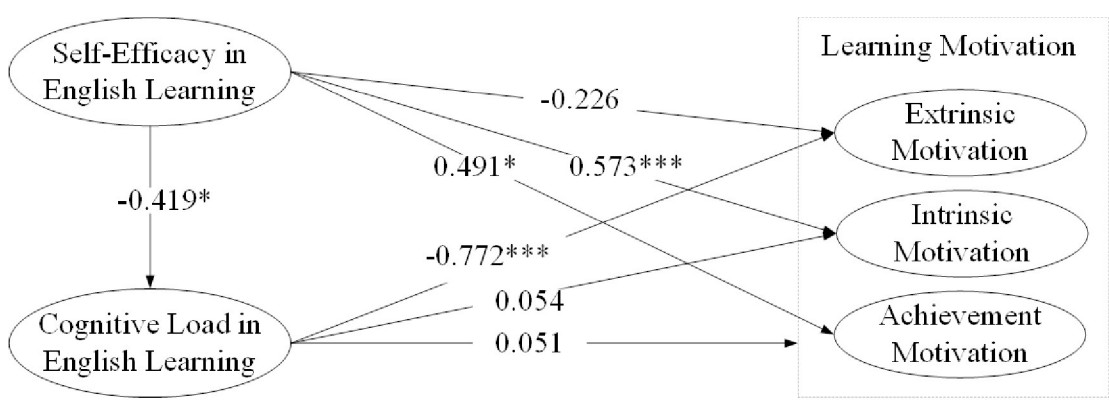

Note: *、**、*** indicate p<0.05, p<0.01, p<0.001 respectively

**Fig 2. Final structural model with estimated parameters.**

motivation ($\beta$ = -0.047, p>0.05) or achievement motivation ($\beta$ = -0.051, p>0.05), so hypothesis 1a, hypothesis H3b, and hypothesis H3c are not supported.

The mediating effect test results are shown in Table 4. Cognitive load significantly mediates the relationship between self-efficacy and extrinsic motivation in English learning, indicating a full mediation effect ($\beta$ = 0.324, p < 0.05). The direct effect of self-efficacy on extrinsic motivation is not significant ($\beta$ = -0.226, p >0.05), suggesting that the influence of self-efficacy on extrinsic motivation is entirely channeled through cognitive load. However, cognitive load does not significantly mediate the relationship between self-efficacy and intrinsic motivation or the relationship between self-efficacy and achievement motivation. In contrast, self-efficacy has a significant direct effect on both intrinsic motivation ($\beta$ = 0.573, p < 0.01) and achievement motivation ($\beta$ = 0.491, p < 0.05)

## Discussion

Our research integrates SCT (SCT) and CLT (CLT) to provide a comprehensive understanding of the dynamics between self-efficacy, cognitive load, and English learning motivation among vocational college students. Our findings not only corroborate the established positive relationship between self-efficacy and learning motivation but also reveal the critical role of cognitive load as a mediator in this relationship. This study extends the existing literature by demonstrating the nuanced interplay between these psychological constructs, particularly in the context of English language learning. The results underscore the importance of self-efficacy in managing cognitive load, thereby enhancing learning motivation, which is a significant contribution to the educational psychology field. The implications of this research should inform the development of pedagogical strategies that effectively nurture learning motivation,

**Table 4. Mediating effect test results.**

| Independent Variable | Intermediate Variable | Dependent Variable | Total Effect | Specific Effect | Direct Effect |
|---|---|---|---|---|---|
| Self-Efficacy | Cognitive Load | Extrinsic Motivation | 0.097 | 0.324* | -0.226 |
| Self-Efficacy | Cognitive Load | Intrinsic Motivation | 0.550*** | -0.023 | 0.573** |
| Self-Efficacy | Cognitive Load | Achievement Motivation | 0.469** | -0.021 | 0.491* |

Note

*, **, *** indicate p<0.05, p<0.01, p<0.001 respectively.

especially in vocational education settings where English language proficiency is crucial for global competitiveness.

## Theoretical implications

This study contributes to the educational psychology literature by integrating Social Cognitive Theory (SCT) and Cognitive Load Theory (CLT) to explore the mediating role of cognitive load in the relationship between self-efficacy and English learning motivation among vocational college students. Our findings resonate with the work of In'am and Sutrisno, who highlighted the role of self-efficacy in stimulating learning motivation through cooperative learning models [72]. Similarly, our research aligns with the insights provided by Shao and Kang, which highlight the pivotal role of self-efficacy in the impact of parent-child relationships on learning engagement [73]. However, unlike previous studies that have often examined self-efficacy and cognitive load in isolation, our study delves into their combined effect, offering a more nuanced understanding of the psychological mechanisms underlying learning motivation. This integration of theories not only broadens the existing theoretical base but also provides a novel framework for educational practice, particularly in enhancing student learning motivation in vocational education settings.

## Practical implications

In the global context of vocational education, where English proficiency is crucial, our research findings offer educators and policymakers targeted strategies to enhance English learning motivation among students: (1) Self-Efficacy Enhancement Develop structured interventions to increase students' self-belief in their capabilities. By designing a series of progressively challenging yet achievable tasks, we can build students' confidence and reinforce self-efficacy, thereby directly boosting intrinsic and achievement motivations. (2) Cognitive Load Management with AI Integration. Apply CLT in lesson planning, focusing on prioritizing essential information and breaking down complex tasks into manageable segments. Integrate AI tools into instruction to personalize learning experiences, adapting the pace and complexity of content to individual students' needs. AI can significantly reduce cognitive load by providing tailored exercises and real-time feedback, creating an efficient learning process that indirectly supports extrinsic motivation. (3) Fostering a Motivational Classroom Environment. Cultivate an environment that nurtures intrinsic motivation by providing students with a clear understanding of the relevance of their learning. Offer constructive feedback and encourage a growth mindset that values effort and persistence. This approach not only fosters a love for learning but also prepares students to embrace challenges and persist in the face of difficulty.

## Conclusion

Our research has contributed to the field of educational psychology by integrating SCT and CLT to provide a more nuanced understanding of the factors influencing English learning motivation among vocational college students. This integration has been empirically validated, revealing the significant role of cognitive load as a mediator between self-efficacy and learning motivation, a relationship that has not been extensively explored in previous literature. Our findings, which are culturally specific to Chinese vocational education, offer valuable insights into the challenges and opportunities faced by students in this rapidly evolving sector. These insights are not only relevant for educators seeking to enhance student motivation but also for researchers interested in the psychological dynamics of learning in diverse educational contexts. Policymakers can leverage our findings to inform the development of educational

strategies that cater to the unique needs of vocational students, thereby ensuring that vocational education remains a vibrant and effective avenue for student success.

In conclusion, our study has shed light on the complex interplay between self-efficacy, cognitive load, and learning motivation, contributing to both theoretical and practical discourses in educational psychology. The insights gained from this research have the potential to inform future studies and educational practices, emphasizing the importance of fostering a supportive learning environment that addresses the cognitive demands of students. As the field of vocational education continues to grow and adapt to global challenges, our work underscores the need for ongoing research and innovation in pedagogical approaches to ensure that students are well-equipped for success in their future endeavors.

## Limitations and future research

This study sheds light on the interplay between cognitive load, self-efficacy, and English learning motivation among vocational college students. However, its cross-sectional design presents limitations. It captures a static view, precluding the determination of causality and the tracking of variable changes over time. The inability to discern whether self-efficacy influences cognitive load and motivation or if these factors are interdependent represents a notable constraint. To overcome this limitation, future research should employ longitudinal methods. Such studies would monitor the evolution of these variables, potentially uncovering the sequence of effects and providing evidence for causality. Additionally, they could assess the influence of external factors like curriculum revisions or policy changes on the dynamics of the constructs studied.

The study's focus on Chinese vocational students also limits the generalizability of the findings. Cross-cultural studies could broaden the analysis, offering insights into the universality of the findings and identifying culture-specific influences on the constructs. This extension would contribute to a more comprehensive understanding of the variables' interplay in diverse educational contexts.

In essence, while this research lays a foundational understanding of the variables' relationships, future longitudinal and cross-cultural studies are crucial for clarifying causal pathways and enhancing the findings' applicability across educational settings.

## Supporting information

**S1 File.**
(DOCX)

## Author Contributions

**Writing – original draft:** Hui Zhang.

**Writing – review & editing:** Hui Zhang.

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
