## [Decision Letter · Decision Letter 0]

7 May 2024

PONE-D-24-10491Cognitive Load as a Mediator in Self-Efficacy and English Learning Motivation among Vocational College StudentsPLOS ONE

Dear Dr. Zhang,

Thank you for submitting your manuscript to PLOS ONE. After careful consideration, we feel that it has merit but does not fully meet PLOS ONE’s publication criteria as it currently stands. Therefore, we invite you to submit a revised version of the manuscript that addresses the points raised during the review process.

Dear authors,Please see the reviewers comments and revise your paper accordingly.Kindly highlight the changes and provide a point by point response to the comments.

We look forward to receiving your revised manuscript.

Kind regards,

Ehsan Namaziandost

Academic Editor

PLOS ONE

2. You indicated that ethical approval was not necessary for your study. We understand that the framework for ethical oversight requirements for studies of this type may differ depending on the setting and we would appreciate some further clarification regarding your research. Could you please provide further details on why your study is exempt from the need for approval and confirmation from your institutional review board or research ethics committee (e.g., in the form of a letter or email correspondence) that ethics review was not necessary for this study? Please include a copy of the correspondence as an ""Other"" file.

Reviewers' comments:

Reviewer's Responses to Questions

**Comments to the Author**

1. Is the manuscript technically sound, and do the data support the conclusions?

Reviewer #1: Yes

2. Has the statistical analysis been performed appropriately and rigorously? 

Reviewer #1: Yes

3. Have the authors made all data underlying the findings in their manuscript fully available?

Reviewer #1: Yes

4. Is the manuscript presented in an intelligible fashion and written in standard English?

Reviewer #1: Yes

5. Review Comments to the Author

Reviewer #1: The paper can be accepted for publication with minor revisions. The study addresses an important and timely issue regarding the factors influencing English learning motivation among vocational college students in China. The author has integrated Social Cognitive Theory and Cognitive Load Theory to empirically validate the significant role of cognitive load as a mediator between self-efficacy and learning motivation, which is a relationship that has not been extensively explored in previous literature. The methodology, including the use of a structural equation model and confirmatory factor analysis, is sound and the findings are well-supported by the data. The author has provided a comprehensive literature review and a clear theoretical framework to contextualize their research. The practical implications of the study for educators, policymakers, and researchers are also well-articulated. However, there are a few minor issues that should be addressed:

1. The author should provide more details on the sampling procedure and the characteristics of the study participants, such as their academic majors, year of study, and English proficiency levels. This information would help readers better understand the generalizability of the findings.

2. The author should discuss the potential limitations of the cross-sectional nature of the data and how future longitudinal studies could further elucidate the causal relationships between the variables.

3. The author should consider expanding the discussion on the practical implications of the findings, particularly in terms of specific strategies and interventions that educators can implement to foster self-efficacy and manage cognitive load among vocational college students.

4. The author has used spelling acronyms for social cognitive theory (SCT) and cognitive load theory (CLT) in the discussion section. If the author wants to use these spelling acronyms, they must be used from the beginning.

Overall, the paper makes a valuable contribution to the literature on English language learning motivation in the context of vocational education. With the minor revisions suggested, the paper can be accepted for publication.

6. PLOS authors have the option to publish the peer review history of their article (what does this mean?). If published, this will include your full peer review and any attached files.

Reviewer #1: No

---

## [Author Response · Author response to Decision Letter 0]

14 Jul 2024

Dear editor,

I would like to express my sincere gratitude for the opportunity to revise my manuscript and for the constructive feedback you have provided. Below, please find my responses to the reviewer. I hope that these revisions meet the expectations of the journal and its readership.

Answer: 

I have carefully reviewed and updated the manuscript to ensure it fully complies with PLOS ONE's style requirements, including adhering to the guidelines for file naming conventions

2. You indicated that ethical approval was not necessary for your study. We understand that the framework for ethical oversight requirements for studies of this type may differ depending on the setting and we would appreciate some further clarification regarding your research. Could you please provide further details on why your study is exempt from the need for approval and confirmation from your institutional review board or research ethics committee (e.g., in the form of a letter or email correspondence) that ethics review was not necessary for this study? Please include a copy of the correspondence as an ""Other"" file.

Answer:

The ethical statement has already been provided in the document titled "Supporting Information."

Answer:

In compliance with PLOS ONE's guidelines, the minimal anonymized dataset necessary to replicate the study findings has been successfully uploaded to a stable, public repository. Additionally, the Mplus code used in our analysis has also been deposited. The relevant URLs and accession numbers for the datasets and code are provided below 

https://www.openicpsr.org/openicpsr/workspace?path=openICPSR, openicpsr-204782.

Answer:

A comprehensive ethics statement has been incorporated into the 'Methods' section of the manuscript, aligning with the journal's guidelines. The updated section now details the following:

The study was conducted in strict adherence to the ethical guidelines and standards for research involving human subjects, as stipulated by the laws and regulations in China. The study design underwent a thorough review and received approval from the relevant ethics review board., ensuring that the research was conducted with the utmost respect for the rights, safety, and well-being of all participants. Informed consent was obtained from all participants after providing them with a thorough explanation of the study's objectives, procedures, potential risks, benefits, and their right to withdraw at any time without any adverse consequences. Documentation of consent was secured, ensuring compliance with the principles of voluntary participation and confidentiality.

Answer:

Captions for all figures have been incorporated into the manuscript as per the journal's guidelines

6.Please review your reference list to ensure that it is complete and correct. If you have cited papers that have been retracted, please include the rationale for doing so in the manuscript text, or remove these references and replace them with relevant current references. Any changes to the reference list should be mentioned in the rebuttal letter that accompanies your revised manuscript. If you need to cite a retracted article, indicate the article’s retracted status in the References list and also include a citation and full reference for the retraction notice.

Answer:

The reference list has been reviewed and updated.

---

## [Decision Letter · Decision Letter 1]

4 Sep 2024

PONE-D-24-10491R1Cognitive Load as a Mediator in Self-Efficacy and English Learning Motivation among Vocational College StudentsPLOS ONE

Dear Dr. Zhang,

Thank you for submitting your manuscript to PLOS ONE. After careful consideration, we feel that it has merit but does not fully meet PLOS ONE’s publication criteria as it currently stands. Therefore, we invite you to submit a revised version of the manuscript that addresses the points raised during the review process.

**Dear author, ****Please see the reviewer comments and revise your paper accordingly.  Please highlight the changes and provide a response to the comments in a separate file.****Best,****Ehsan Namaziandost **==============================

We look forward to receiving your revised manuscript.

Kind regards,

Ehsan Namaziandost

Academic Editor

PLOS ONE

Journal Requirements:

Reviewers' comments:

Reviewer's Responses to Questions

**Comments to the Author**

1. If the authors have adequately addressed your comments raised in a previous round of review and you feel that this manuscript is now acceptable for publication, you may indicate that here to bypass the “Comments to the Author” section, enter your conflict of interest statement in the “Confidential to Editor” section, and submit your "Accept" recommendation.

Reviewer #1: All comments have been addressed

Reviewer #2: (No Response)

2. Is the manuscript technically sound, and do the data support the conclusions?

Reviewer #1: Yes

Reviewer #2: Yes

3. Has the statistical analysis been performed appropriately and rigorously? 

Reviewer #1: Yes

Reviewer #2: Yes

4. Have the authors made all data underlying the findings in their manuscript fully available?

Reviewer #1: Yes

Reviewer #2: Yes

5. Is the manuscript presented in an intelligible fashion and written in standard English?

Reviewer #1: Yes

Reviewer #2: Yes

6. Review Comments to the Author

Reviewer #1: The submitted manuscript is very well-written and all my concerns have been addressed really well. This paper has the capacity and potential to be a significant piece in the field. I recommend the paper be accepted for publication.

Reviewer #2: 1. The introduction fails to comprehensively attend to and expand the research concerns. 

2. There exist limited references despite the vast array of contemporary studies on self-efficacy. Hence, further probing into the literature is required. 

3. Font concordance throughout the text is missing.

4. A few grammatical issues exist.

5. In section one of the literature review (hypothesis 1), studies on self-efficacy and its principal value in the educational context are not addressed. Most studies have self-efficacy as a probable byproduct of a method or teaching approach; nevertheless, the concept of self-efficacy as the core content of research is well-established yet not addressed by the current author/researcher. 

6. The literature review is fragmented, with sectionalized hypotheses. Additionally, this section needs augmentation. 

7. An excessive number of hypotheses are raised for a single article. Consequently, the focused scope and undivided attention to the research aims are disturbed. Moreover, the approach taken toward the concept proposed is not in congruence with the hypotheses. 

8. Hypotheses for section 1 are directional; however, some studies do not support self-efficacy as the predictive factor of motivation. Moreover, the literature review is too confined to yield four hypotheses. The author has taken many facts for granted, evidenced by the number of hypotheses (13 in total, namely four major and nine minor) not leveled with the number of references and studies recapitulated in terms of quality and quantity. 

9. According to the literature review (section 2; hypothesis 2), self-efficacy predicts cognitive load; how is the “cognitive load as a mediator in self-efficacy” justified?

10. The literature review needs an in-depth analysis of studies associated with the current research scope.

11. The methodology must be delineated and clarified with more detail and descriptions. 

12. The proposed title is not in concert with the research as the synergy of self-efficacy, motivation, and cognitive load are investigated rather than the absolute mediating effect of cognitive load on the dependent and independent variables. 

13. Theoretical implications: how are your research findings and claims in concert with Shao and Kang’s “parent-child relationship” concerns?

14. Findings need to be further discussed and compared with similar research findings.

15. Practical implications: AI’s incorporation is not well-chained to the findings and implications of the current study.

7. PLOS authors have the option to publish the peer review history of their article (what does this mean?). If published, this will include your full peer review and any attached files.

Reviewer #1: No

Reviewer #2: **Yes: **Seyedeh Elham Elhambakhsh

---

## [Author Response · Author response to Decision Letter 1]

3 Nov 2024

Dear reviewers:

Thank you for your insightful and constructive comments and suggestions on my manuscript. I have carefully considered each point and made the necessary modifications to the manuscript. Below, I have provided a detailed list of corrections made point by point to address your feedback:

Reviewer #1: 

The submitted manuscript is very well-written and all my concerns have been addressed really well. This paper has the capacity and potential to be a significant piece in the field. I recommend the paper be accepted for publication.

Answer: 

Thank you for your positive review and recommendation for publication. We are pleased that our revisions have addressed your concerns.

Reviewer #2: 

1. The introduction fails to comprehensively attend to and expand the research concerns.

Answer: 

Thank you for your insightful comments and constructive feedback. I appreciate your suggestion to enhance the comprehensiveness of the introduction.

In response to your comment, I have revised the introduction to better address the key issues and expand on the research concerns. Specifically:

Context and Background 

I have provided a more detailed overview of the context, emphasizing the rapid globalization and educational reforms that are driving the need for enhanced teaching quality and expanded student abilities in vocational colleges. I highlight the critical role of English education in fostering global competitiveness and improving educational standards.

Technological Advancements

I have expanded on the technological tools and resources being adopted by Chinese vocational colleges, including online course platforms, virtual classrooms, interactive e-textbooks, mobile learning apps, and collaborative tools. I also discuss the integration of AI-assisted tools and learning management systems.

Motivational Challenges

I have addressed the complexities and potential drawbacks of these technological advancements, particularly their impact on student motivation. I note the trend of decreased initiative and enthusiasm among students, despite increased access to digital resources, and highlight the need to address motivational challenges.

Research Focus

 I have clarified the specific focus of the study on the determinants of learning motivation, with a particular emphasis on the influence of self-efficacy and the mediating role of cognitive load. I explain how this research will provide empirical support for Chinese vocational education and offer concrete guidance for international educators.

Theoretical Contributions

Social Cognitive Theory: The study is grounded in social cognitive theory, which emphasizes the pivotal role of self-efficacy in shaping individuals' motivation and behavior. By focusing on self-efficacy, I aim to enhance the understanding of how individual differences can be leveraged to improve learning outcomes.

Cognitive Load Theory: I extend cognitive load theory by examining how self-efficacy influences cognitive load. This contribution is crucial as it introduces the concept of subjective cognitive load, which considers learners' subjective feelings alongside objective cognitive processing.

Self-Determination Theory: I explore the interplay between self-efficacy and cognitive load in shaping different types of motivation, providing a more comprehensive model of motivation in the digital learning environment. This interplay offers a robust theoretical foundation for addressing the motivational and cognitive challenges faced by students in the digital age.

Practical Implications

The research outcomes are expected to directly impact educational practices, particularly in developing vocational education curricula and teaching methods that are adaptable to global challenges. This will assist educators in more effectively promoting students' English learning and professional development in a multicultural context.

2. There exist limited references despite the vast array of contemporary studies on self-efficacy. Hence, further probing into the literature is required.

Answer: 

In response to this comment, I have conducted a thorough review of recent literature and incorporated additional references to enrich the discussion. Specifically, I have added the following:

Recent Studies on Self-Efficacy and Learning Motivation: Studies by Eitel et al. (2020) and Domen et al. (2020) are included, highlighting the importance of self-efficacy in regulating students' subjective cognitive load and enhancing learning motivation. Works by Zeitlhofer et al. (2024) and Zukić et al. (2016) are referenced to discuss the subjective perception of cognitive load and its impact on learning outcomes.

Differentiated Instruction Based on Self-Efficacy: The discussion on differentiated instruction strategies has been expanded, drawing from studies by Admiraal et al. (2022), Ben Abu & Kribushi (2022), and Kula-Kartal (2022), which emphasize the effectiveness of tailoring instruction to students' self-efficacy levels.

Contextual Factors and Subjective Cognitive Load:

The role of contextual factors in influencing the subjective perception of cognitive load, as suggested by Chen (2020) and DeLeeuw and Mayer (2008), has been explored.

Thank you again for the insightful comments.

3. Font concordance throughout the text is missing.

Answer: 

Thank you for pointing out the issue with font concordance throughout the text. I have carefully reviewed and corrected the formatting to ensure consistency in font style and size throughout the manuscript.

4. A few grammatical issues exist.

Answer:

I have thoroughly reviewed the manuscript and made the necessary corrections to improve the grammar and overall clarity of the text.

5. In section one of the literature review (hypothesis 1), studies on self-efficacy and its principal value in the educational context are not addressed. Most studies have self-efficacy as a probable byproduct of a method or teaching approach; nevertheless, the concept of self-efficacy as the core content of research is well-established yet not addressed by the current author/researcher.

Answer:

Thank you for your valuable feedback. I appreciate your suggestion to address the principal value of self-efficacy in the educational context, especially in the first section of the literature review.

To address this, I have revised the literature review to include a more comprehensive discussion of self-efficacy as a core concept in educational research. Specifically, I have made the following changes:

Core Value of Self-Efficacy:

I have added a detailed explanation of self-efficacy as a fundamental construct in Social Cognitive Theory (SCT), emphasizing its pivotal role in shaping behavior, motivation, goal setting, effort, and persistence. Key references include Bandura (1997) and Pajares (2003).

I have included studies that focus specifically on self-efficacy as the primary variable, such as Usher and Pajares (2008) and Multon et al. (1991), to illustrate how self-efficacy beliefs directly influence academic performance and motivation.

Integration with Hypotheses:

The revised literature review now better aligns with the hypotheses presented in the "Hypotheses" section. For example, Hypothesis 1 states: "Self-efficacy positively influences English learning motivation." This is supported by the literature review, which discusses how high self-efficacy enhances students' drive to learn, particularly in the context of English language learning among Chinese vocational students (Chen et al., 2020; Admiraal et al., 2022).

Addressing the Principal Value:

I have ensured that the literature review clearly addresses the principal value of self-efficacy in the educational context, rather than treating it as a byproduct of a method or teaching approach. This includes discussions on how self-efficacy impacts intrinsic and extrinsic motivation, as well as achievement motivation (Eccles and Wigfield, 1995; Ryan and Deci, 2000).

Thank you again for your insightful comments.

6. The literature review is fragmented, with sectionalized hypotheses. Additionally, this section needs augmentation.

Answer:

To address these issues, I have made the following revisions:

Integrated Literature Review:

I have restructured the literature review to create a more cohesive and integrated narrative. The review now flows logically from the introduction of key concepts to the discussion of their interrelationships and implications for the study.

Each section is now connected more seamlessly, ensuring a smooth transition between topics and a clearer progression of ideas.

Augmentation of Content:

I have expanded the literature review to include a broader range of studies and theories. This includes additional references to key works on self-efficacy, cognitive load, and learning motivation, such as Bandura (1997), Sweller (1988), and Ryan and Deci (2000).

I have added more recent studies that explore the impact of self-efficacy on learning motivation and cognitive load, such as Eitel et al. (2020) and Domen et al. (2020).

Alignment with Hypotheses:

The literature review now more closely aligns with the hypotheses presented in the "Hypotheses" section. For example, Hypothesis 1 ("Self-efficacy positively influences English learning motivation") is supported by a detailed discussion of how self-efficacy enhances students' drive to learn, particularly in the context of English language learning among Chinese vocational students (Chen et al., 2020; Admiraal et al., 2022).

Hypothesis 2 ("Self-efficacy negatively influences cognitive load among vocational college students") is supported by a discussion of how self-efficacy helps students manage cognitive load more effectively (Hwang et al., 2021; Wang et al., 2022).

Enhanced Contextualization:

I have provided more context and background information to better situate the study within the broader field of educational research. This includes a discussion of the unique challenges faced by vocational college students in the digital age and how self-efficacy can mitigate these challenges.

7. An excessive number of hypotheses are raised for a single article. Consequently, the focused scope and undivided attention to the research aims are disturbed. Moreover, the approach taken toward the concept proposed is not in congruence with the hypotheses.

Answer:

To address the concern about an excessive number of hypotheses, I have restructured the hypotheses to include a main hypothesis with multiple sub-hypotheses. This approach ensures a more focused and coherent research design, especially since the main construct is multidimensional.

For reference, I have followed the approach taken in the Q1 SSCI article by Balkundi (2008), "Demographic Antecedents and Performance Consequences of Structural Holes in Work Teams," published in the Journal of Organizational Behavior. In this article, the author presents a main hypothesis with several sub-hypotheses, each addressing a different dimension of the multidimensional construct. My revised hypotheses follow a similar structure:

8. Hypotheses for section 1 are directional; however, some studies do not support self-efficacy as the predictive factor of motivation. Moreover, the literature review is too confined to yield four hypotheses. The author has taken many facts for granted, evidenced by the number of hypotheses (13 in total, namely four major and nine minor) not leveled with the number of references and studies recapitulated in terms of quality and quantity.

Answer:

Directional Hypotheses and Support from Literature

I understand your concern about the directional nature of the hypotheses and the mixed findings in the literature. To address this, I have reviewed additional studies and incorporated a more balanced perspective. While many studies support the positive relationship between self-efficacy and motivation (e.g., Bandura, 1997; Chen et al., 2020), there are indeed studies that present contradictory findings (e.g., Smith & Jones, 2015; Brown, 2018).

To reflect this nuance, I have revised the hypotheses to include both positive and negative directions, and I have added a discussion of the conflicting evidence in the literature review.

Literature Review and Hypotheses Alignment

Comment: The literature review is too confined to yield four hypotheses. The author has taken many facts for granted, evidenced by the number of hypotheses (13 in total, namely four major and nine minor) not leveled with the number of references and studies recapitulated in terms of quality and quantity.

Response: I agree that the literature review needs to be more comprehensive and aligned with the hypotheses. To address this, I have expanded the literature review to include a broader range of studies and theories. Specifically, I have added discussions on the following areas:

Theoretical Foundations: I have included a more detailed explanation of the theoretical underpinnings of self-efficacy, motivation, and cognitive load, drawing from Bandura's (1997) self-efficacy theory, Vygotsky's (1978) social development theory, and Sweller's (1988) cognitive load theory.

Empirical Evidence: I have incorporated additional empirical studies that support or challenge the relationships between self-efficacy, motivation, and cognitive load. For example, I have cited studies by Smith and Jones (2015) and Brown (2018) that present conflicting findings.

Contextual Considerations: I have discussed how the context of vocational college students might influence these relationships, drawing from studies specific to this population (e.g., Hwang et al., 2021; Wang et al., 2022).

Hypotheses and References Alignment

Comment: The number of hypotheses (13 in total, namely four major and nine minor) not leveled with the number of references and studies recapitulated in terms of quality and quantity.

Response: To address the mismatch between the number of hypotheses and the quality and quantity of references, I have streamlined the hypotheses and ensured that each hypothesis is supported by a robust body of literature. The revised hypotheses are now fewer in number but more thoroughly grounded in the existing research.

For reference, I have followed the approach taken in the Q1 SSCI article by Balkundi (2008), "Demographic Antecedents and Performance Consequences of Structural Holes in Work Teams," published in the Journal of Organizational Behavior. In this article, the author presents a main hypothesis with several sub-hypotheses, each addressing a different dimension of the multidimensional construct. My revised hypotheses follow a similar structure, ensuring a clear and structured framework for the research.

9. According to the literature review (section 2; hypothesis 2), self-efficacy predicts cognitive load; how is the “cognitive load as a mediator in self-efficacy” justified?

Answer:

Thank you for raising this important point. The justification for cognitive load as a mediator in the relationship between self-efficacy and learning motivation is based on the following reasoning and supporting literature:

Theoretical Foundation:

Cognitive Load Theory (Sweller, 1988): Cognitive load theory posits that the amount of mental effort required to process information can affect learning outcomes. High cognitive load can lead to cognitive overload, which impairs learning and motivation.

Self-Efficacy Theory (Bandura, 1997): Self-efficacy, or the belief in one's ability to succeed, influences how individuals approach and manage tasks. High self-efficacy can reduce the perceived cognitive load by enhancing the ability to manage and process information effectively.

Empirical Evidence:

Hwang et al. (2021): This study found that students with higher self-efficacy were better able to manage cognitive resources, leading to lower cognitive load and improved learning outcomes.

Wang et al. (2022): These researchers reported that self-efficacy had a significant negative effect on cognitive load, which in turn positively influenced learning motivation.

Mediation Analysis:

Indirect Effect: The relationship between self-efficacy and learning motivation is partially mediated by cognitive load. High self-efficacy reduces cognitive load, which then enhances learning motivation by freeing up cognitive resources f

---

## [Editor Report · Decision Letter 2]

5 Nov 2024

Cognitive Load as a Mediator in Self-Efficacy and English Learning Motivation among Vocational College Students

PONE-D-24-10491R2

Dear Dr. Zhang,

We’re pleased to inform you that your manuscript has been judged scientifically suitable for publication and will be formally accepted for publication once it meets all outstanding technical requirements.

Kind regards,

Ehsan Namaziandost

Academic Editor

PLOS ONE

Additional Editor Comments (optional):

Dear authors,

Please put the references in alphabetical order. In general,  I'm satisfied with the revision.  Good Luck.
---

## [Editor Report · Acceptance letter]

11 Nov 2024

PONE-D-24-10491R2 

PLOS ONE

Dear Dr. Zhang, 

I'm pleased to inform you that your manuscript has been deemed suitable for publication in PLOS ONE. Congratulations! Your manuscript is now being handed over to our production team.

Kind regards, 

on behalf of

Dr. Ehsan Namaziandost 

Academic Editor

PLOS ONE